# Trust in healthcare providers, information sources, and concerns for new maternal vaccines among pregnant and lactating women in Kenya

Jessica L. Schue[1], Berhaun Fesshaye[1], Emily Miller[1], Prachi Singh[1], Molly Sauer[1], Rosemary Njogu[2], Rose Jalang'o[3], Joyce Nyiro[4], Ruth A. Karron[5], Rupali J. Limaye[1,6]*

1 Department of International Health, International Vaccine Access Center, Johns Hopkins Bloomberg School of Public Health, Baltimore, Maryland, United States of America, 2 Jhpiego, Nairobi, Kenya, 3 National Vaccines and Immunization Program, Ministry of Health Kenya, Nairobi, Kenya, 4 KEMRI-Wellcome Trust, Kilifi, Kenya, 5 Department of International Health, Johns Hopkins Bloomberg School of Public Health, Baltimore, Maryland, United States of America, 6 Department of Global and Community Health, George Mason College of Public Health, Fairfax, Virginia, United States of America

* rlimaye@jhu.edu

## Abstract

New maternal vaccines have the potential to reduce morbidity and mortality for infants from common illnesses that pose the greatest risk in the earliest phase of their life. Respiratory syncytial virus (RSV) is a leading cause of acute lower respiratory infections among infants under six months of age. With the recent approval of a maternal vaccine for RSV, this study aimed to understand decision-making factors among pregnant and lactating women for receiving a newly licensed vaccine during pregnancy. Pregnant and lactating women from two counties in Kenya, Nakuru and Mombasa, were recruited to complete a cross-sectional survey in July-September 2022. The survey explored topics of trust in various types of sources for information about new maternal vaccines, the importance of a healthcare provider's recommendation of a new maternal vaccine, and concerns about new maternal vaccines. We surveyed 400 pregnant and lactating women. In both counties, information about the new vaccine was most trusted when coming from healthcare providers, and least trusted when coming from social media. Women's intention to receive a new maternal vaccine was heavily influenced by a positive recommendation from a healthcare provider. The greatest concerns about a new vaccine were side effects and the vaccine's ingredients. The information and recommendation from a healthcare provider are important influences on decision-making for new maternal vaccines. As a new maternal immunization for RSV becomes more available, healthcare providers should be engaged early to reduce vaccine hesitancy amongst providers and equip providers with appropriate information tailored to pregnant women about the RSV maternal vaccine.

**Data availability statement:** We cannot share data as there is potentially identifying information that could identify participants. All data requests can be sent to either of the following IRB contacts: KEMRI: director@kemri.org JHU IRB: BSPH.irboffice@jhu.edu.

**Funding:** This study was funded by the Gates Foundation (grant INV-016977) to R.J.L/R.A.K. at the Johns Hopkins Bloomberg School of Public Health. The funders had no role in study design, data collection and analysis, decision to publish, or preparation of the manuscript.

**Competing interests:** The authors have declared that no competing interests exist.

## Introduction

Maternal immunization, or immunization during pregnancy, allows the pregnant person to pass antibodies through the placenta to the fetus, thereby providing passive immunity that protects infants at birth and in the first weeks and months of life [1]. Vaccination during pregnancy can also serve to protect the pregnant person from experiencing severe disease or death themselves, and maternal infections are themselves a contributor to preterm birth which can subsequently contribute to neonatal mortality [2,3].

The first vaccine to be administered during pregnancy was tetanus toxoid to prevent neonatal tetanus [4]. Since then, tetanus-diphtheria (Td) or tetanus-diphtheria-pertussis (Tdap) and SARS-CoV2/COVID-19 vaccines have been added to the list of maternal immunizations that are now commonly implemented worldwide [4]. In high- and middle-income countries, influenza and, most recently, respiratory syncytial virus (RSV) vaccines are recommended for pregnant women, whereas these vaccines are not standard in low- and middle-income countries [5,6]. Looking ahead, several maternal Group B Streptococcus (GBS) vaccine candidates are currently in development to help overcome the limitations of existing prevention strategies and better protect newborns from infection [7], and maternal immunization against malaria and tuberculosis is also being considered.

Respiratory syncytial virus (RSV) is the leading cause of acute lower respiratory infections in children under five, and in 2019, was estimated to cause more than 33 million infections, 3.6 million hospital admissions, and more than 100,000 deaths [8]. Low- and middle-income countries (LMICs) bear a disproportionate burden of the disease, accounting for 95% of infections and 97% of deaths [8]. Additionally, RSV disease is most severe when children are entering their first RSV season, and preterm infants are disproportionately affected, making up 25% of RSV hospitalization burden in 2019 [1,9].

A bivalent RSVpreF maternal vaccine has been approved for use and recommended to pregnant women in more than 40 high- and middle-income countries as of the end of 2024, with the World Health Organization recommending that all countries introduce either the maternal vaccine or a long-acting monoclonal antibody to protect infants against RSV disease [10,11]. Unlike maternal RSV vaccines, long-acting RSV monoclonal antibodies are administered to infants to provide protection for several months, but are costly with extremely limited supply, leaving them out of reach in all but higher-income settings [12,13]. Given the limited availability of RSV monoclonal antibodies in LMICs, the maternal RSV vaccine is particularly important. The RSV maternal vaccine is recommended for use during the third trimester of pregnancy, with recommendations for the exact gestational age windows varying by country [10,14].

Previous research in Kenya has shown a substantial burden of RSV-related illness and death among children under 5 years of age, with highest levels of morbidity and mortality occurring in the first 6 months of life [15]. Additionally, research on seasonality has found that maximum temperature, absolute humidity, and weekly number

of births are drivers of annual RSV epidemics in Kenya [16]. While non-medically attended RSV disease accounts for most of the country's burden, RSV hospitalization is also a significant cause of economic burden for households in Kenya [17,18]. This makes preventative care like vaccination in pregnancy especially important as it can not only decrease RSV-related morbidity and mortality, but also decrease economic burden in Kenya. There has been limited prior research done specifically with pregnant women in Kenya on demand generation and information sources for a maternal RSV vaccine. One study of pregnant and lactating women and their community members found that healthcare workers were trusted advisors and information sources, as were family and community members and traditional media [19]. Other studies with pregnant and lactating women and Kenyan healthcare workers found that most of the participants were not aware of RSV disease or maternal vaccines and recommended the creation of RSV awareness education materials [20,21]. Additionally, a systematic review that included studies conducted in Kenya found that attitudes toward RSV prevention products were relatively positive despite low knowledge of RSV disease, similarly emphasizing the importance of education on RSV disease and prevention products for all target audiences, includingpregnant women, healthcare providers, and policymakers [22].

Given that a maternal RSV vaccine may be available in Kenya in the next few years and the importance of effectively educating and communicating about this vaccine, this study aims to describe which information sources pregnant women trusted, their attitudes toward vaccines in pregnancy, and their intentions to vaccinate as future maternal vaccines become available.

## Methods

The target population of this study was pregnant and lactating women in both rural and urban settings of Kenya. To this end, participants were recruited from 20 health facilities in Nakuru County (rural) and Mombasa County (urban); these counties were identified in consultation with Ministry of Health officials and in-country partners to balance population and geographic diversity with accessibility and feasibility for data collection. A range of health facility types was included to improve the representativeness of the sample, from level 2 (community clinics) through level 5 (specialized referral hospitals). We enrolled 400 pregnant or lactating women, assuming an α of 0.05, 5% precision, and a conservative estimate of 50% prevalence of vaccine hesitance among pregnant women, given the variability of coverage data on maternal vaccination in Kenya; we allowed for 3% incompletion or non-response and rounded our sample size up slightly for more even distribution between the two counties. We evenly distributed the target sample between the two counties, with facility allocations varying by catchment population and typical facility attendance.Further details on sample size and facilities are described by Limaye et al [23].

Data collectors completed a standard training, which included topics related to the ethical interaction with human participants (role of the data collector, voluntary participation, informed consent, personal privacy, working with vulnerable populations, protection of personal information), data integrity (collection, reporting, and storing data), as well as instrument practice including iterative feedback. The survey instrument was pretested and revised as needed based on language and tone. Scenarios were also provided to data collectors related to these topics to ensure competency of topics and participant interactions.

Recruitment procedures differed by facility; generally, participants were approached consecutively upon arrival at antenatal clinics, maternity wards, or maternal and child health units. If a participant met the inclusion criteria (18+, currently breastfeeding or in the second or third trimester of pregnancy, and able to consent), oral consent was obtained, with a place on the consent form for the participant to mark an X to indicate their consent. Surveys were administered in either English or Swahili using tablets. Data collection occurred from 25/07/2022 through 21/09/2022. This study received ethical approval from the Kenya Medical Research Institute Scientific and Ethics Review Unit (protocol 4211) in Kenya and Johns Hopkins Bloomberg School of Public Health Institutional Review Board (study IRB00014893) in the US.

The survey instrument included sociodemographic questions, trust in information sources, the importance of a healthcare provider's recommendation, and concerns about new maternal vaccines. The survey started with questions about RSV risk perception and specific questions about an RSV maternal vaccine, followed by questions about new maternal vaccines broadly. Constructs included in the questionnaire were based on and adapted from relevant theories and models pertaining to vaccine acceptance and behaviors and validated survey tools. These included the SAGE vaccine hesitancy survey tool development guidance, 5C psychological antecedents of vaccination, behavioral and social drivers of vaccination (BeSD) model, socio-ecological model, and health belief model [24–28], as well as the validated Parent Attitudes about Childhood Vaccines (PACV) questionnaire [29–31] and WHO SAGE Vaccine Hesitancy Scale (VHS) [32]. We modified survey items from these tools to focus on maternal vaccination; where appropriate, we separated questions to inquire about mother and newborn separately. The survey instrument was further refined following review and validation during the data collector training.

*Trust in information sources.* We asked women to indicate their level of trust in a variety of sources, asking participants: "I trust the information that I have received from xx about vaccines during pregnancy." We asked about the following sources: health care provider, social media platforms, scientists and doctors at universities and academic institutions, and media (TV, radio), and women were given a 4-point Likert scale for answer choices (strongly agree, agree, disagree, strongly disagree).

*Healthcare provider recommendation.* We asked women to indicate their likelihood of receiving a new vaccine based upon their healthcare provider's recommendation. We asked: "If a new vaccine was approved for pregnant women, and your health care provider did not recommend it, how likely would you get the new vaccine?" and "If a new vaccine was approved for pregnant women, and your health care provider did recommend it, how likely would you get the new vaccine?". Women were given a 4-point Likert scale for answer choices (very likely, likely, unlikely, very unlikely).

*New vaccine concerns.* We asked a question about future vaccine concerns and participants ranked their answers from 1-7 in order of concern, with 1 being most concerning and 7 being least concerning: "When a new vaccine is approved for use and is recommended for me/family, I am typically concerned with the following: a) ingredients in the vaccine; b) side effects; c) availability of the vaccine at my health facility; d) costs to get the vaccine; e) what others are saying about the vaccine; f) healthcare providers recommendation to get the vaccine; and g) a family member, such as a partner or mother-in-law's input about me getting the vaccine."

## Data analysis

Data were stratified by county and summary statistics of binary and categorical data (including Likert scale data) were calculated for each variable of interest. Categorical demographic data were compared by county using Pearson's chi-squared test of independence. The level of trust participants had in information sources was summarized and compared by county. Concerns regarding new vaccines and the influence of a health care provider recommendation were summarized by calculating the percent of respondents that chose a concern for each rank.

## Results

We surveyed 400 participants, evenly split between Nakuru and Mombasa counties. The majority of participants were between the ages of 18–29, had at least one child, and more than half of participants had a secondary education or higher (Table 1).

In both Nakuru and Mombasa, 98% of respondents agreed or strongly agreed that they would trust information about new maternal vaccines in pregnancy provided by healthcare providers. A majority of respondents in both counties also trusted information from scientists and/or doctors and media, including TV and radio. Trust in information from social media was low in both counties, as 78% and 80% for Mombasa and Nakuru respectively disagreed that they could trust social media (Fig 1).

**Table 1. Sociodemographic characteristics of study sample.**

| | *Nakuru (N = 200)* | *Mombasa (N = 200)* | *p-value[1]* |
|---|---|---|---|
| Age, n(%) | | | |
| *18-29* | 132 (66.0) | 137 (68.5) | |
| *30-44* | 68 (34.0) | 63 (31.5) | 0.594 |
| Lactating, n(%) | | | |
| *Pregnant* | 51 (25.5) | 50 (25.0) | |
| *Lactating* | 149 (74.5) | 150 (75.0) | 0.908 |
| Number of children under 18, n(%) | | | |
| *None* | 19 (9.5) | 20 (10.0) | |
| *One* | 63 (31.5) | 69 (34.5) | |
| *Two* | 66 (33.0) | 58 (29.0) | |
| *Three* | 28 (14.0) | 36 (18.0) | |
| *Four or more* | 24 (12.0) | 17 (8.5) | 0.556 |
| Education level, n(%) | | | |
| *Less than primary school* | 15 (7.5) | 30 (15.0) | |
| *Primary school completed* | 68 (34.0) | 62 (31.0) | |
| *Secondary/ High school completed* | 66 (33.0) | 55 (27.5) | |
| *College/University completed* | 51 (25.5) | 53 (26.5) | 0.097 |

[1] p-values are from Pearson's Chi-square test of independence

We then asked participants about healthcare provider recommendations. If their healthcare provider recommended a new maternal vaccine, 63% of women in Nakuru and 70% of women in Mombasa indicated that they were very likely to get the new maternal vaccine. If their healthcare provider did not recommend a new maternal vaccine, 5% of women in each county indicated that they were very likely to get the new maternal vaccine (Fig 2).

Related to concerns about new maternal vaccines, we examined how participants ranked concerns by county of residence and displayed these in a heatmap. The concern that was most frequently ranked as the most important across both counties was vaccine side effects, with the second concern being the vaccine's ingredients. In both counties, opinions of family members and others were the lowest ranked concerns. (Fig 3).

## Discussion

Healthcare providers were highly trusted sources for information about new maternal vaccines among pregnant and lactating women in both Nakuru and Mombasa counties in Kenya. A healthcare provider's recommendation to receive or not receive a new maternal vaccine holds considerable influence on women as they decide whether to be vaccinated in pregnancy. The main concerns about a new maternal vaccine are the vaccine's side effects and ingredients. Participants were least concerned about their family's and others' opinions about them getting a new maternal vaccine. Responses were similar between the two counties, suggesting negligible difference in opinions between urban and rural populations. While these specific responses were about new maternal vaccines broadly, the questions were prompted by describing maternal RSV vaccines. We believe that this description likely influenced participants to answer questions within the context of maternal RSV vaccines.

While research on maternal RSV vaccines in low- and middle-income countries is lacking, there have been a few studies focused on knowledge and acceptance of RSV vaccines in pregnancy, but these have been conducted in primarily high-income settings. For example, a study among Australian women focused on RSV and Group B Streptococcus (GBS) vaccines in pregnancy found that safety data was the most important factor for acceptance [33]. These findings align

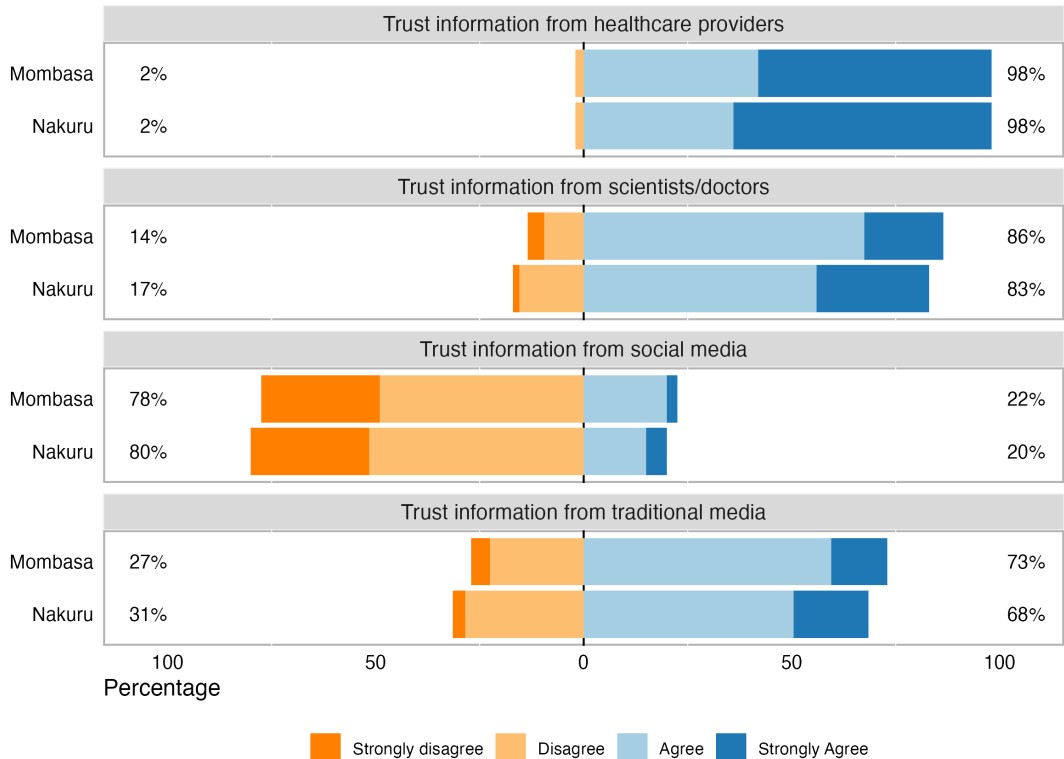

**Fig 1. Level of trust in information about new maternal vaccines by source.** No respondents chose Strongly Disagree in response to trusting information from healthcare providers. Percentages may not equal 100% due to rounding.

with ours as side effects, an important measure for safety, was the biggest concern among women in both counties. For another example, a sample of women in Jordan also found favorable opinions about RSV vaccination during pregnancy, given the vaccine is safe [34]. The concern of vaccine ingredients, which emerged as the second most important in both counties, was not as prevalent in other studies. More research is needed to better understand RSV knowledge and vaccine acceptance among women living in low- and middle-income countries to inform vaccine demand efforts.

Previous research, including from Kenya, has shown that healthcare providers are important and highly trusted sources for health information, including vaccination in pregnancy [35–37]. Aligned with our findings, a recommendation for vaccination from a healthcare provider is central to vaccine acceptance, and the absence of a recommendation can even decrease uptake [35,36]. One study also found that pregnant women even preferred to speak with healthcare providers directly about vaccination, instead of receiving printed communication materials [38]. Similar to our results, two previous studies in Kenya also found that traditional media sources such as TV and radio, were utilized and trusted more than social media sources [39,40]. However, despite the potential presence of misinformation on social media, some studies have found social media and Internet platforms as important sources of health information for mothers [41,42]. This points to the importance of ensuring accurate information is present on social media to increase trust and present correct information to those that are already relying on these platforms. Understanding whom women may go to and seek information related to the maternal RSV vaccine will be crucial to equip trusted messengers with the right information once vaccine demand efforts begin.

The growing evidence base on maternal vaccination acceptance highlights the importance of targeted and tailored communication strategies, particularly those that equip healthcare providers to deliver clear and consistent information to

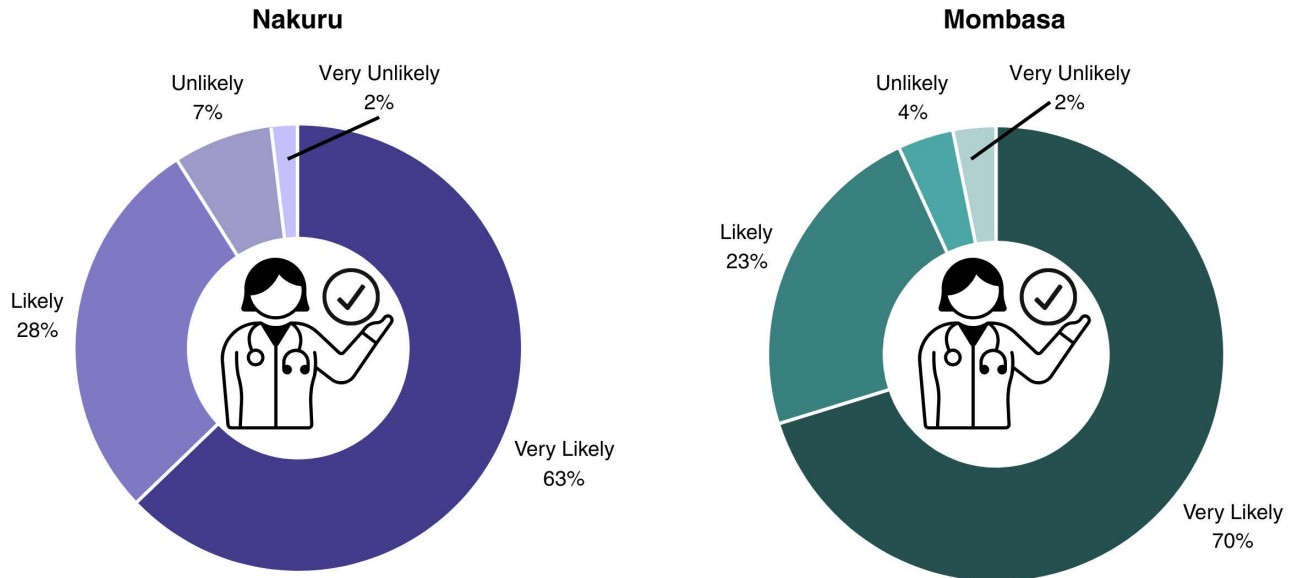

Likely to get a new maternal vaccine if healthcare provider did recommend

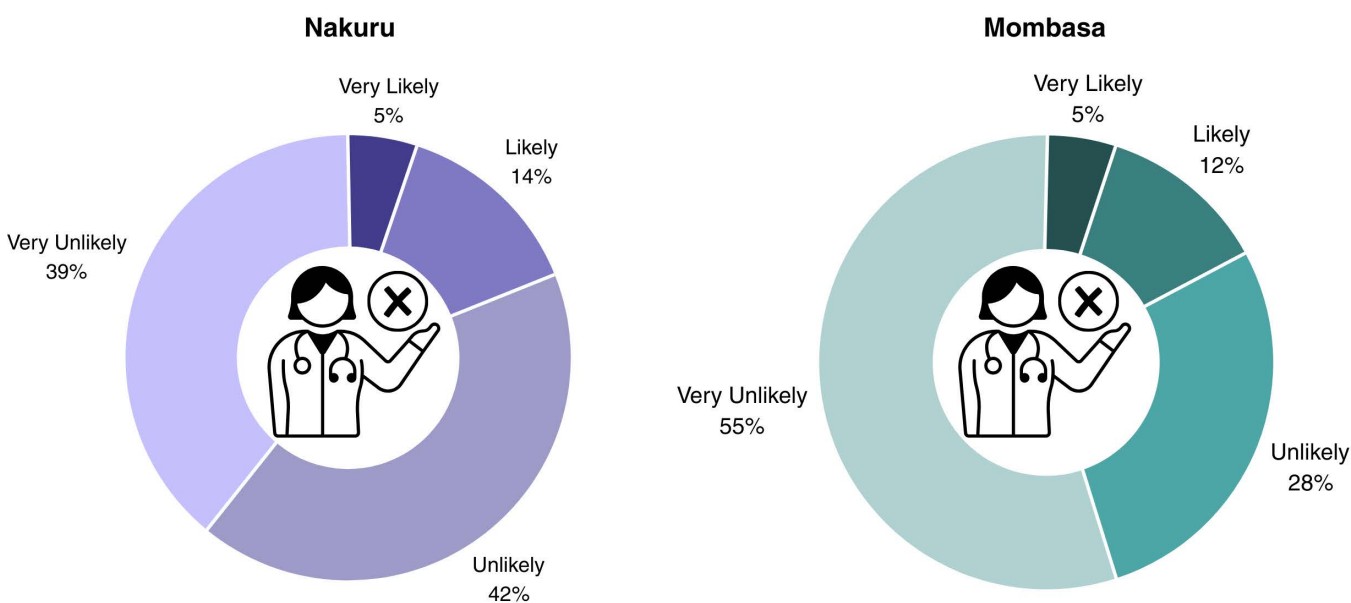

Likely to get a new maternal vaccine if healthcare provider did not recommend

**Fig 2. Influence of healthcare provider recommendations on maternal vaccine acceptance.**

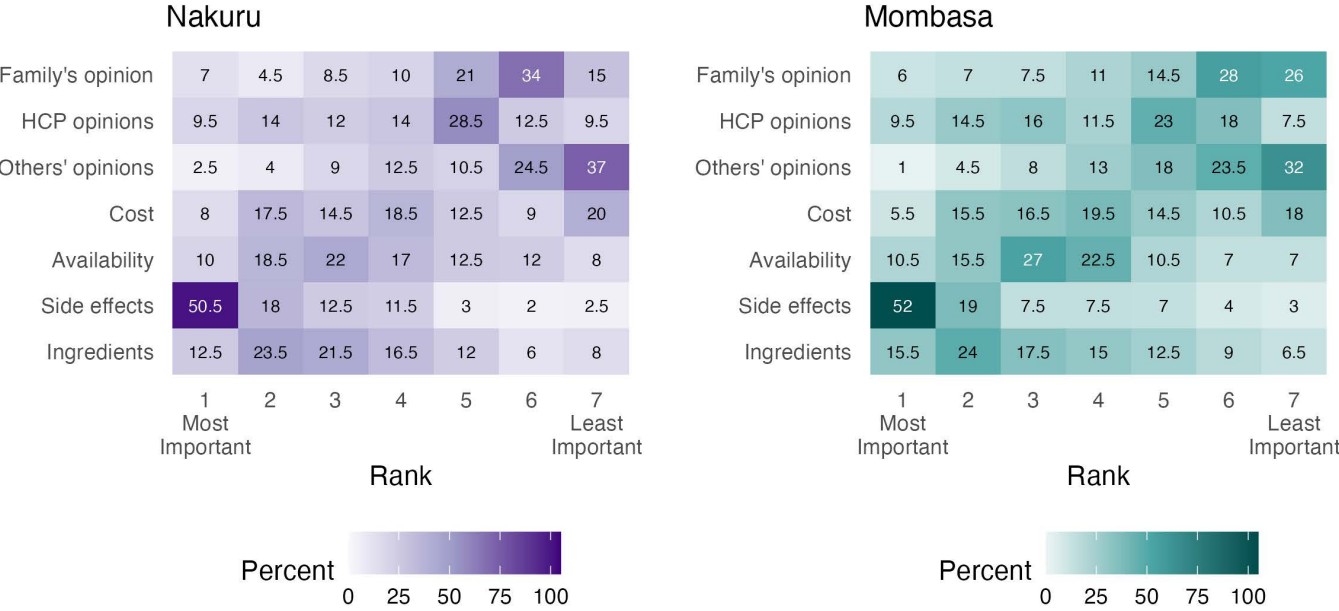

**Fig 3. Concerns about new maternal vaccines ranked.** Values and shades of tiles represent the percent of respondents that chose a particular concern at each rank level. HCP: Healthcare provider.

their patients throughout pregnancy. Safety concerns, like side effects or concerns about vaccine ingredients, are commonly identified in similar studies. Strengthening surveillance systems to monitor and evaluate adverse events among pregnant women and newborns is one strategy to improve the information available to providers and program officials to guide recommendations and healthcare provider recommendations [43]. Distinguishing serious AEFI from local vaccine reactions, and understanding how common they are, may also help build trust and mitigate concerns about side effects and ingredients. For example, relatively minor side effects (i.e., pain at the injection site, swelling, etc.) from tetanus vaccination in pregnancy have not deterred vaccine acceptance [44]. Discussing these experiences may help providers assuage patients' concerns and foster greater acceptance of future maternal vaccines. As safety is the most pressing concern related to a new maternal RSV vaccine, ensuring that the framing, tone, and delivery of communication used to address safety concerns is nuanced and persuasive will be required for successful rollout.

RSV vaccine is recommended in the third trimester, a relatively narrow window compared to other vaccines in pregnancy (i.e., tetanus toxoid) which may be given at any time in pregnancy. A study in Kenya found that two in three pregnant women had their first antenatal care (ANC) visit prior to this window [45]. This may lay the groundwork for healthcare providers to prepare pregnant women for newer maternal vaccines by providing information and addressing concerns during their first ANC visits, before they are eligible to receive these vaccines. For example, Nyawanda et al. noted that healthcare providers in Kenya describe suboptimal RSV disease knowledge but are highly supportive of RSV vaccination for pregnant women [21]. This highlights the importance of strengthening healthcare provider training and knowledge in advance of vaccine availability, including interpersonal communication techniques to engage women early in their pregnancies and support informed RSV vaccine decision-making and follow-up.

Our study found that media is a trusted information source among pregnant and lactating women; this mirrors the findings of a systematic review by De Brabandere and colleagues, which elucidated the importance of the internet and social media in shaping maternal vaccine knowledge and attitudes [41]. Targeted media and social media campaigns in advance of RSV vaccine availability should include messages emphasizing the opportunity for mothers to protect their newborns by

being vaccinated themselves, and including information about vaccine safety as well as efficacy. As the RSV vaccine has been primarily rolled out in higher-income countries, studies in Australia and the U.S. have emphasized the effectiveness of media and social media messaging that centers the dual benefits of maternal vaccination for both mother and baby, and the unique agency of mothers to further protect their newborns through maternal vaccination [46,47].

This study had several limitations. First, participants were exclusively recruited from health facilities, so the opinions of women who do not seek care through the formal health system are missed. We sought to recruit a diverse sample of study participants related to age, education, and parity. In addition, social desirability bias may have influenced participants to respond with more favorable views of vaccines or healthcare providers than they actually believed. We adapted established, validated vaccine hesitancy survey tools for our study but note that there may be important differences when inquiring about vaccination in pregnancy; further study to validate this adapted scale would be beneficial. Despite these limitations, the study also had notable strengths. Participants were recruited from both urban and rural settings in different parts of the country. This is also one of the first studies to address information needs for future maternal vaccines, which is crucial for demand generation and community sensitization.

Pregnant and lactating women rely on trusted sources to inform their decision-making related to maternal vaccines. As our study revealed that a strong recommendation from a healthcare provider would likely be the most important influence, it will be imperative to equip healthcare providers with salient and digestible information to be able to strongly recommend a future maternal RSV vaccine. Healthcare providers can also be vaccine hesitant, and efforts to provide relevant information to them well before a maternal RSV vaccine is available is therefore important to realize the potential benefits of a maternal RSV vaccine at the population level.

## Supporting information

**S1 File. Inclusivity in global research.**
(DOCX)

## Author contributions

**Conceptualization:** Ruth A. Karron, Rupali J. Limaye.

**Formal analysis:** Jessica L. Schue, Rupali J. Limaye.

**Funding acquisition:** Ruth A. Karron, Rupali J. Limaye.

**Investigation:** Rosemary Njogu, Ruth A. Karron, Rupali J. Limaye.

**Methodology:** Rupali J. Limaye.

**Project administration:** Berhaun Fesshaye, Emily Miller, Prachi Singh, Rosemary Njogu, Ruth A. Karron, Rupali J. Limaye.

**Resources:** Rosemary Njogu, Ruth A. Karron, Rupali J. Limaye.

**Software:** Rupali J Limaye.

**Supervision:** Rosemary Njogu, Ruth A. Karron, Rupali J. Limaye.

**Validation:** Rupali J. Limaye.

**Visualization:** Jessica L. Schue, Berhaun Fesshaye, Emily Miller, Prachi Singh, Molly Sauer.

**Writing – original draft:** Jessica L. Schue, Berhaun Fesshaye, Emily Miller, Prachi Singh, Molly Sauer, Ruth A. Karron, Rupali J. Limaye.

**Writing – review & editing:** Jessica L. Schue, Berhaun Fesshaye, Emily Miller, Prachi Singh, Molly Sauer, Rosemary Njogu, Rose Jalang'o, Joyce Nyiro, Ruth A. Karron, Rupali J. Limaye.

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
