## [Decision Letter · Decision Letter 0]

20 May 2025

PGPH-D-25-00691

Trust in healthcare providers, information sources, and concerns for new maternal vaccines among pregnant and lactating women in Kenya

Dear Dr. Limaye,

Thank you for submitting your manuscript to PLOS Global Public Health. After careful consideration, we feel that it has merit but does not fully meet PLOS Global Public Health’s publication criteria as it currently stands. Therefore, we invite you to submit a revised version of the manuscript that addresses the points raised during the review process.

EDITOR: 

Ensure that the manuscript conforms to the reporting guidelines for an observational study (STROBE).

We look forward to receiving your revised manuscript.

Kind regards,

Claire E. von Mollendorf

Academic Editor

Journal Requirements:

2. In the online submission form, you indicated that [Data will be made available upon reasonable request.].

a. In a public repository,

b. Within the manuscript itself, or

c. Uploaded as supplementary information.

Additional Editor Comments (if provided):

Reviewers' comments:

Reviewer's Responses to Questions

**Comments to the Author**

1. Does this manuscript meet PLOS Global Public Health’s publication criteria?

Reviewer #1: Yes

Reviewer #2: Yes

2. Has the statistical analysis been performed appropriately and rigorously?

Reviewer #1: I don't know

Reviewer #2: Yes

3. Have the authors made all data underlying the findings in their manuscript fully available (please refer to the Data Availability Statement at the start of the manuscript PDF file)?

Reviewer #1: Yes

Reviewer #2: Yes

4. Is the manuscript presented in an intelligible fashion and written in standard English?

Reviewer #1: Yes

Reviewer #2: Yes

Reviewer #1: I appreciate the opportunity to review your study aiming to understand decision-making factors among pregnant and lactating women for receiving a newly licensed vaccine during pregnancy. Below are my comments and suggestions that I believe would enhance the quality and impact of your research.

Major Points

Introduction:

• It is imperative for the authors to cite and discuss more than one prior research study addressing the same research problem. This context will underscore the novelty of your study and help differentiate it from previous attempts.

• Clearly state the aim and objectives of your study at the conclusion of the introduction section to provide a concise overview of your research goals.

Methods:

• The sampling strategy should be clearly defined and reported in the methods section.

• Authors should clarify how the sample size was determined.

• Line 110 “Data collectors completed a standard training, and the survey instrument was pretested.” What does ‘standard’ mean? What did they do in the training and what made you sure they were ready to collect data?

• The authors fail to provide any information in their manuscript that would allow readers to understand how the sample differs from the target population, particularly in terms of key sociodemographic characteristics such as age, gender, and others. This lack of information raises concerns about the representativeness of the sample.

• Authors should clarify how they estimated the reliability or internal consistency of the questionnaire used, using, for example, Cronbach’s alpha to measure whether a score is reliable.

• More information should be provided regarding the considerations concerning the literature review (please add the references) and the authors' previous research experience in developing the questionnaire.

• The study should use additional statistical analysis. Specifically, information on how the authors assessed the normality of numeric variables and any relevant assumptions made for the chosen statistical tests or models should be reported to enhance the transparency and reliability of the statistical analysis. Also, information should be added regarding the logistic regression analysis used (e.g., independent and dependent variables, assumptions, etc.)

• Line 139-140 “New vaccine concerns. We asked question about future vaccine concerns and asked 140 participants to rank their answers from 1-6,…”. Not clear whether the ranking is a score or the number of options, if number of options 7 were actually given.

Results:

• Detailed findings from the statistical analyses used such as the logistic regression models should be reported in the text of the Results section. The authors should create new tables reporting the results from these analyses, which should be included among the main tables of the manuscript.

Discussion:

• The discussion section does well to explore the underlying meaning of the research and its potential implications in other areas of study. The authors have discussed how their findings contribute to the existing knowledge and highlight the significance of their research within a broader context. However, they should consider addressing possible improvements or future directions that can further develop the concerns of their research.

• Lines 195-197 “While these specific responses were about new maternal vaccines broadly, the questions were prompted by describing maternal RSV vaccines.” Critically state if you feel this made any difference.

• Lines 199-208 “While research on maternal RSV vaccines in low- and middle-income countries is lacking, there have been a few studies focused on knowledge and acceptance of RSV vaccines in pregnancy. A study among Australian women focused on RSV and Group B Streptococcus (GBS) vaccines in pregnancy found that safety data was the most important factor for acceptance (17). These findings align with ours as side effects, an important measure for safety, was the biggest concern among women in both counties. A sample of women in Jordan also found favorable opinions about RSV vaccination during pregnancy, given the vaccine is safe (18). The concern of vaccine ingredients, which emerged as the second most important in both counties, was not as prevalent in other studies.”. You begin the paragraph talking about low- and middle-income countries but go on to discuss Australia and Jordan instead.

• Lines 240-251 “RSV vaccine is recommended in the third trimester, a relatively narrow window 241 compared to other vaccines in pregnancy (i.e., tetanus toxoid) which may be given at 242 any time in pregnancy. A study in Kenya found that two in three pregnant women had 243 their first antenatal care (ANC) visit prior to this window (29). This may lay the 12 244 groundwork for healthcare providers to prepare pregnant women for newer maternal 245 vaccines by providing information and addressing concerns during their first ANC visits, 246 before they are eligible to receive these vaccines. Our study found that media is a 247 trusted information source among pregnant and lactating women. Targeted media and 248 social media campaigns in advance of RSV vaccine availability should include 249 messages emphasizing the opportunity for mothers to protect their newborns by being 250 vaccinated themselves, and including information about vaccine safety as well as 251 efficacy (30).” You begin the paragraph talking about uptake of RSV vaccine and then sharply switch to discussion about the media. These should be two separate paragraphs expanded on and discussing other published research.

Reviewer #2: This cross-sectional survey aimed to understand decision-making factors among pregnant and lactating women for receiving a newly licensed vaccine during pregnancy. As far as respiratory syncytial virus (RSV) is a leading cause of acute lower respiratory infections among infants under six months of age, it is important to conduct cross-sectional survey to understand which information sources pregnant women trusted and their intentions toward future maternal vaccines given in pregnancy, and conducting survey will have an integral role in the working process of determining appropriate measures required to equip healthcare providers with salient and digestible information to be able to strongly recommend a future maternal RSV vaccine, and pregnant and lactating women rely on trusted sources to inform their decision-making related to maternal vaccines.

The manuscript is technically sound, and the data supports the conclusions, and it describes methodologically and ethically rigorous research with conclusions that are appropriately drawn based on the data presented. Furthermore, conclusions are presented in an appropriate fashion and are supported by the data, and the data presented in the manuscript supported the conclusions drawn, and the authors did not overstate their conclusions. The language in the submitted article is clear, correct, and unambiguous and is written in standard English. The research meets all applicable standards for the ethics of experimentation and research integrity. The article adheres to appropriate reporting guidelines and community standards for data availability. Moreover, the manuscript is well organized and written clearly enough to be accessible to non-specialists.

Generally, the article is suitable for publication subject to the following minor corrections or comments for revision and submission to current or other journals. Here below I tried to put my concerns!

Abstract:

Dear author, in the abstract section, you stated that pregnant and lactating women from two counties in Kenya, Nakuru and Mombasa, were recruited to complete a cross-sectional survey in August-September 2022. However, in the methods section, you stated that data collection occurred from 25/07/2022 through 21/09/2022. It is recommended that the author should report consistently.

Another concern in the abstract section: it is better to include an adjusted odds ratio (AOR) with a 95% confidence interval (CI) which was used to measure the strength of the association between factors and the outcome variable. In addition, it is better to include the statistical software used for data entry and analysis and the sampling methods used to recruit pregnant and lactating women from two counties in Kenya, Nakuru and Mombasa, to complete a cross-sectional survey.

Methods:

Dear author, You stated that participants were recruited from 20 health facilities in Nakuru County (rural) and Mombasa County (urban). A range of health facility types was included, from level 2 through level 5 (line numbers 107-109). However, you did not provide in detail the sampling technique and procedure employed to arrive at the desired sample size (400 pregnant and lactating women).

Furthermore, it is recommended to provide the sample size determination, data processing and analysis used in this study.

Results:

Dear author, You reported that nearly half of participants had at least a secondary education (line number 150). However, Table 1 showed that the percentages of participants who completed a secondary education in Nakuru and Mombasa counties were 33% and 27.5%, respectively. It is better to revise it.

Regarding trust in information from traditional media, 31% disagreed and 68% agreed for Nakuru participants (Figure 1). The summation of participants who disagreed and agreed was 99%. Dear author, the summation of participants should be 100%. It is recommended that the author should revise Figure 1.

**Do you want your identity to be public for this peer review?** For information about this choice, including consent withdrawal, please see our Privacy Policy

Reviewer #1: No

Reviewer #2: No

---

## [Decision Letter · Decision Letter 1]

20 Oct 2025

Trust in healthcare providers, information sources, and concerns for new maternal vaccines among pregnant and lactating women in Kenya

PGPH-D-25-00691R1

Dear Dr. Limaye,

We are pleased to inform you that your manuscript 'Trust in healthcare providers, information sources, and concerns for new maternal vaccines among pregnant and lactating women in Kenya' has been provisionally accepted for publication in PLOS Global Public Health.

Best regards,

Claire E. von Mollendorf

Academic Editor

Reviewer Comments (if any, and for reference):

Reviewer's Responses to Questions

**Comments to the Author**

Reviewer #2: All comments have been addressed

publication criteria?

Reviewer #2: Yes

3. Has the statistical analysis been performed appropriately and rigorously?

Reviewer #2: Yes

4. Have the authors made all data underlying the findings in their manuscript fully available (please refer to the Data Availability Statement at the start of the manuscript PDF file)?

Reviewer #2: Yes

5. Is the manuscript presented in an intelligible fashion and written in standard English?

Reviewer #2: Yes

Reviewer #2: Dear Authors,

Thank you for your diligent efforts in addressing the peer-review comments. Your revisions have significantly enhanced the manuscript’s clarity and quality. The dates in the methods section have been corrected to align with those in the abstract. You have also provided additional details on sample size determination, sampling strategy, and data analysis as recommended.

Moreover, the results section now clearly indicates that over half of the participants had secondary education or higher, encompassing both secondary and college-level education. Your thorough responses to the peer-review report and adherence to the manuscript revision criteria are appreciated. Each point has been carefully considered and necessary adjustments have been made to improve the overall quality and clarity of the manuscript.

However, it would be beneficial to include a limitations section at line 329 and ensure the conclusion section starts at line 343. Please ensure these sections are added.

Overall, the article is suitable for publication.

**Do you want your identity to be public for this peer review?** For information about this choice, including consent withdrawal, please see our Privacy Policy

Reviewer #2: **Yes: ** Gemechu Dereje Feyissa
